# Scaffold Chemical Model Based on Collagen—Methyl Methacrylate Graft Copolymers

**DOI:** 10.3390/polym15122618

**Published:** 2023-06-08

**Authors:** Yulia L. Kuznetsova, Ksenya S. Gushchina, Karina S. Lobanova, Victoria O. Chasova, Marfa N. Egorikhina, Alexandra O. Grigoreva, Yulia B. Malysheva, Daria A. Kuzmina, Ekaterina A. Farafontova, Daria D. Linkova, Yulia P. Rubtsova, Luydmila L. Semenycheva

**Affiliations:** 1Faculty of Chemistry, National Research Lobachevsky State University of Nizhny Novgorod, 23, Gagarin Ave., 603022 Nizhny Novgorod, Russia; kyul@yandex.ru (Y.L.K.); ksesha.gushchina@gmail.com (K.S.G.); kariandrs2101@yandex.ru (K.S.L.); 11.04.96@mail.ru (V.O.C.); yu.b.malysheva@gmail.com (Y.B.M.); kuzminarabota@yandex.ru (D.A.K.); 2Federal State Budgetary Educational Institution of Higher Education, Privolzhsky Research Medical University of the Ministry of Health of the Russian Federation, 603005 Nizhny Novgorod, Russia; egorihina.marfa@yandex.ru (M.N.E.); ekaterina_farafontova@mail.ru (E.A.F.); linckovadaria@yandex.ru (D.D.L.); rubincherry@yandex.ru (Y.P.R.)

**Keywords:** collagen, poly(methyl methacrylate), tributylborane, quinone, radical polymerization, grafted copolymer

## Abstract

Polymerization of methyl methacrylate (MMA) in aqueous collagen (Col) dispersion was studied in the presence of tributylborane (TBB) and *p*-quinone: 2,5-di-tert-butyl-*p*-benzoquinone (2,5-DTBQ), *p*-benzoquinone (BQ), duroquinone (DQ), and *p*-naphthoquinone (NQ). It was found that this system leads to the formation of a grafted cross-linked copolymer. The inhibitory effect of *p*-quinone determines the amount of unreacted monomer, homopolymer, and percentage of grafted poly(methyl methacrylate) (PMMA). The synthesis combines two approaches to form a grafted copolymer with a cross-linked structure—“grafting to” and “grafting from”. The resulting products exhibit biodegradation under the action of enzymes, do not have toxicity, and demonstrate a stimulating effect on cell growth. At the same time, the denaturation of collagen occurring at elevated temperatures does not impair the characteristics of copolymers. These results allow us to present the research as a scaffold chemical model. Comparison of the properties of the obtained copolymers helps to determine the optimal method for the synthesis of scaffold precursors—synthesis of a collagen and poly(methyl methacrylate) copolymer at 60 °C in a 1% acetic acid dispersion of fish collagen with a mass ratio of the components collagen:MMA:TBB:2,5-DTBQ equal to 1:1:0.015:0.25.

## 1. Introduction

In modern medicine, bioengineered structures involving cellular technologies are applied to restore damaged tissues. The task of creating a good tissue equivalent is to find the right balance between creating conditions for the regeneration of this tissue and restoring its main functions [1]. The composition of such a structure includes three necessary components: scaffold, signaling factors, and cells, commonly referred to as the triad of tissue engineering [2]. There are specific requirements for scaffolds:The scaffold should be biodegradable so that no additional surgical intervention is required to remove the implant, and, in an ideal case, the degradation rate should correspond to the formation rate of new tissue [3,4];The scaffold and its decomposition products should be biocompatible and not cause particular mutagenic, carcinogenic, or cytotoxic effects [4,5,6];The qualities of the implanted scaffold should correspond to the properties of the recipient’s natural tissue [7];The scaffold must have an interconnected network of pores, which will ensure the possibility of its colonization by cells and circulation of gases and liquids and, ultimately, causes tissue growth in all directions [2,8];The scaffold should have properties that ensure the attachment, migration, and proliferation of cells [9];The scaffold should have optimal bioengineering qualities that will allow the material to restore damaged tissues and organs, simulating physiological processes in the replaced biostructures [10,11,12].

For the formation of most tissues when creating scaffolds, natural (collagen, chitosan, alginate, agarose, fibrin, fibronectin) and synthetic (poly-α-hydroxyesters, in particular polylactic acid, polyglycolic acid, their mixture, polyethylene glycol) polymers or hydrogels formed by covalent or ionic cross-linking of water-soluble natural and synthetic polymers are used [13,14,15,16].

The disadvantages of synthetic scaffolds are the lack of cellular recognition and sometimes low biocompatibility. Biopolymers such as collagen or fibronectin have numerous advantages in tissue engineering, such as low toxicity, low immunogenicity, and biodegradability. Natural polymers are preferable in forming the scaffold because sections of molecules capable of binding to cell receptors are preserved. However, their use as scaffolds is often complicated because they do not have sufficient strength and elasticity and have a fibrillar rather than three-dimensional structure [17,18,19,20]. Natural polymers are given that structure by copolymerization with other natural [21,22,23,24,25,26] or synthetic polymers [27,28,29,30,31] to eliminate mentioned disadvantages.

Organoborane compounds are used to synthesize grafted copolymers [32,33,34]. Tributyl borane makes it possible to obtain copolymers of collagen and poly(methyl methacrylate) [32,33], triethyl borane–copolymer of collagen and poly(butyl acrylate) [34], and the combination of triethyl borane and *p*–quinone—a copolymer of starch and poly(methyl acrylate) [35]. At the same time, alkylborane in the composition of various initiators makes it possible to obtain grafted copolymers both by “grafting from” [32,33,34] or “grafting to” [35]. Under certain conditions, the grafted copolymer can be obtained by combing both approaches. In this case, the resulting product must have a cross-linked structure required in scaffold technologies. This was first implemented in our work [36], which involved the synthesis of gelatin and methyl methacrylate copolymers in the presence of tributylborane and 2,5-di-tert-butyl-*p*-benzoquinone. In this case, the chain begins to grow on the borated collagen (“grafting from”) and breaks off on the borated collagen (“grafting to”).

This work aimed to create a chemical model of scaffold construction based on graft-copolymers of collagen and methyl methacrylate using tributyl borane and *p*-quinones. For this, it was necessary to complete the following tasks:Obtain copolymers of methyl methacrylate and fish collagen in the presence of tributyl borane and several *p*-quinones differing in their structure and reactivity, in addition to the previously obtained data [36];Characterize the obtained products using the methods of gel-penetrating chromatography and scanning electron microscopy;Evaluate the propensity of copolymers for biodegradation under the action of enzymes in comparison with collagen to understand the prospects of their use as precursors of scaffolds;Conduct a cytotoxicity examination;Form a chemical model of scaffold construction based on collagen and methyl methacrylate graft-copolymers using tributyl borane and *p*-quinones.

## 2. Materials and Methods

### 2.1. Materials

Collagen dispersion in acetic acid (Mn = 244 kDa, Mw = 279 kDa, Mz = 304 kDa, and PDI = 1.14) was obtained using the process described in [37]. The methods used to purify organic solvents and MMA were outlined in [38]. NQ (Sigma-Aldrich, St. Louis, MI, USA) and DQ (Sigma-Aldrich, St. Louis, MI, USA) were used without preliminary purification. BQ and 2,5-DTBQ were synthesized by oxidation of the corresponding hydroquinone. All *p*-quinones were purified via recrystallization from petroleum ether. The purity of all substances used was monitored by NMR spectroscopy.

### 2.2. Synthesis of TBB

TBB was synthesized by reacting BF_3_∙Et_2_O with 1-butyl bromide over magnesium shavings in diethyl ether by the method outlined in [39]. ^11^B NMR (128 MHz, CDCl_3_) was conducted with a value of δ 86.7.

### 2.3. Polymerization Procedure

Thirty mL of 1% Col solution was placed in a three-necked flask equipped with a mechanical stirrer and reflux condenser, and it was heated in a water bath to 60 °C in an argon atmosphere. A solution of TBB in heptane was placed in a vacuum ampoule. The required TBB volume (0.08 g (0.015 mol.%)) was taken with an argon-filled syringe and poured into a collagen solution with intensive stirring. The mixture was kept for 30 min. After that, a degassed solution containing 0.3 mL of MMA and 0.25 (0.15, 0.35, 2.5) mol.% of *p*-quinone (2,5-DTBQ, DQ, BQ, and NQ) was added to the reaction flask. The reaction mixture was kept for 3 h.

### 2.4. Determination of the Unreacted Monomer

Unreacted MMA was measured via Knopp bromination as described in [39].

MAA concentration was determined by:(1)$$X,\%=\frac{\left(a-b\right)\;\times \;M}{200-m},$$

a, b—volumes of Na_2_S_2_O_3_ expended on the probe and blank titration, respectively, mL; M—MAA molecular weight (MW), g/mole; m—copolymer probe mass, g.

### 2.5. Enzymatic Hydrolysis

The enzymatic hydrolysis of copolymers was carried out using collagenase, thrombin (Renam NPO, Moscow, Russia), and pancreatin (Hubei Maxpharm Industries Co, Ltd., Wuhan, China) with proteolytic activity of 2 U/mg.

The hydrolysis of collagenase was implemented in this work as outlined in [36].

For thrombin and pancreatin hydrolysis, 1 M NaOH was added to the solution to neutralize any acid, with the solution subsequently brought to the required volume with distilled water. The hydrolysis was carried out by adding the enzyme to the resulting mixture at a mass ratio of copolymer to an enzyme of 10^3^:1. Samples (1 mL) were taken at regular intervals (1, 10, 30, 60 min, and 3 days) after the addition of the enzyme. To interrupt the hydrolysis, 1 mL of 4% acetic acid solution was added to the samples.

The mixtures from each hydrolysis were filtered out. The remaining PMMA was concentrated and analyzed using size-exclusion chromatography (SEC).

### 2.6. Size-Exclusion Chromatography

The aqueous dispersion of Col and PMMA copolymer was analyzed on an LC-20 HPLC system (Shimadzu, Kyoto, Japan) with a low-temperature light-scattering detector ELSD-LT II with the LC-Solutions-GPC software module. Measurements were performed at the following conditions: the column—Tosoh Bioscience TSKgel G3000_SWxl_ (Tosoh, Tokyo, Japan) with a 5.0 μm pore size; column temperature—30 °C; the eluent—0.5 M acetic acid solution with the 0.8 mL/min flow rate. Narrow disperse dextran standards with an MW range of 1–410 kDa (Fluka, Buchs, Switzerland) were used for calibration.

Graft-PMMA driven out by enzymatic hydrolysis from a copolymer was analyzed on a Prominence LC-20VP system (Shimadzu, Kyoto, Japan) with the following conditions: column—Tosoh Bioscience (polystyrene-divinylbenzene gel, 10^6^ and 10^5^ Å pore size) (Tosoh, Tokyo, Japan); column temperature—40 °C; eluent—tetrahydrofuran with a 0.7 mL/min flow rate. A differential refractometer and a UV detector (λ = 254 nm) were used as a detector. Calibration was performed using narrow dispersion of PMMA standards.

### 2.7. Scanning Electron Microscopy

The surface of collagen and copolymer samples was studied using a scanning electron microscope (SEM) JSM-IT300LV (JEOL Ltd., Akishima, Japan) with an electron probe diameter of 4 nm (operating voltage 30 kV) by using detectors of low-energy secondary electrons in a low-vacuum mode to prevent the samples from charging. The sponges for the electron microscope were obtained via freeze-drying. The pore size of freeze-dried samples was determined from micrographs using a scale bar.

### 2.8. NMR Spectroscopy

^1^H NMR spectra were recorded on Agilent DD2 400 instrument. Chemical shifts (*δ*) are given in ppm relative to the solvent reference as an internal standard (*δ* 7.26 ppm for CDCl_3_).

### 2.9. Cytotoxicity Examination via MTT Assay

Human dermal fibroblasts (HDF) of 4–6 passages were used to study the samples. An active, morphologically homogeneous culture was used, which adheres well to plastic. The immunophenotype of the culture cells corresponded to the immunophenotype of mesenchymal cells. The sustainability of the culture was more than 97%. HDF with a density of 10 thousand per 1 cm^2^ was seeded into the wells of a flat-bottomed 96-well tablet in DMEM/F12 growth medium, with antibiotics (penicillin, 100 units/mL; streptomycin, 100 μg/mL) and 10% inactivated calf embryonic serum. Then, they were cultivated under standard conditions for one day. After 24 h, even growth of typical fibroblast-like cells in the form of a subconfluent monolayer was recorded in all wells. After 24 h of cultivation, the growth medium above the cells was replaced with the studied solutions.

The MTT assay is a colorimetric method based on the reaction of the recovery of the tetrazolium dye 3-(4,5-dimethylthiazole-2-yl)-2,5-diphenyl-tetrazolium bromide into insoluble formazan. The formazan concentration reflected in the optical density was measured using a Sunrise tablet reader (Austria) at a wavelength of 540 nm.

The test samples in the form of films brought to a constant mass in a vacuum cabinet were placed in a prepared growth medium (DMEM/F12 with antibiotics and 2% calf embryonic serum). Samples were placed in a CO_2_ incubator for 24 h under standard conditions (temperature—37 °C; CO_2_ concentration—5%). After 24 h of incubation, the extract was taken from the studied samples, and a series of dilutions with the growth medium was prepared in the ratios 1:1, 1:2, 1:4, and 1:8.

Various concentrations of the extract were added to the culture of HDF sown in a 96-well tablet. Each was performed in 8 dexterities (8 holes). The sample tablet was placed in a CO_2_ incubator for 72 h.

After 72 h, 20 μL of a solution of MTT (5 mg/mL in a phosphate-buffered solution) was added to each well. Then, the cells were incubated with MTT for 3 h in a CO_2_ incubator. After 3 h of incubation, the supernatant was selected and replaced with an equal volume of DMSO solution. The optical density (OD) was recorded at 540 nm on a tablet reader.

The relative growth intensity (RGI) was calculated according to the following formula:(2)$$\mathrm{RGI},\%=\frac{\mathrm{average}\;\mathrm{OD}\;\mathrm{in}\;\mathrm{test}\;\mathrm{culture}}{\mathrm{average}\;\mathrm{OD}\;\mathrm{in}\;\mathrm{control}}\times 100.$$

The RGI of the experimental series was compared with the RGI of the control, taking it to 100%.

The severity of cytotoxicity was ranked as follows:RGI 100%—rank 0;RGI 99–75%—rank 1 corresponds to the absence of cytotoxicity;RGI 74–50%—rank 2 corresponds to a mild degree of cytotoxicity;RGI 49–25%—rank 3 corresponds to an average degree of cytotoxicity;RGI 24–0%—rank 4 corresponds to high cytotoxicity [40].

## 3. Results and Discussion

To create a chemical model of scaffold construction based on collagen and methyl methacrylate copolymers using tributyl borane and *p*-quinones, it was necessary, in addition to the previously obtained data [33,36,39], to synthesize copolymers in the presence of tributyl borane and some *p*-quinones differing in their structure and reactivity, and evaluate their parameters as materials for scaffolds. As in previously published studies, copolymers were synthesized under varying temperature conditions—25 °C, 45 °C, and 60 °C. It is known [41] that fish collagen undergoes denaturation and gradually turns into its analog—gelatin. The properties of gelatin in comparison with collagen in catalytic processes (enzymatic catalysis of protein hydrolysis) and its functional properties in hybrid hydrogel scaffolds differ slightly [42,43,44,45,46]. In this regard, we conducted research including a temperature of synthesis at 60 °C when part of the collagen can turn into gelatin. It was taken into consideration when interpreting the results.

For this research, collagen–PMMA copolymers (Col–*co*–PMMA) were synthesized using the following *p*-quinones: 2,5-di-tert-butyl-*p*-benzoquinone 
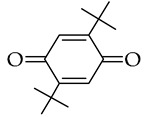
, *p*-benzoquinone 
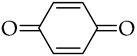
, duroquinone 
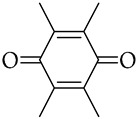
, and *p*-naphthoquinone 
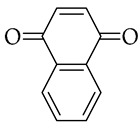
.

The obtained material was studied in detail in the example of 2,5-DTBQ. The mass ratio of collagen and MMA used for synthesis is 1:1. The amount of TBB is equivalent to the number of hydroxyproline units contained in collagen because it was shown in previous works [33,36] that the maximum ratio of grafted MMA was due to the boration reaction of the hydroxyl group of hydroxyproline. The concentration of 2,5-DTBQ in MMA is 0.25 mol.%, which was commonly used in the synthesis of (co)polymers in the presence of the alkylborane–quinone system [47]. As shown in Table 1, the conversion of the monomer does not depend much on temperature and is slightly higher at a synthesis temperature of 60 °C (calculated from unreacted monomer).

The molecular weight (MW) determined by the GPC method and the morphology of freeze-dried samples were determined for the copolymers. In addition, enzymatic hydrolysis of copolymer samples with pancreatin and thrombin enzymes was carried out, with molecular weight characteristics controlled during the process. The results of hydrolysis were compared with those for collagen. It was not possible to estimate the MW parameters of the formed copolymer because the molecular weight distribution (MWD) curves of the samples (Figure 1, curves 2–4) differ little from the MWD curves of collagen (Figure 1, curve 1). We suppose, when preparing highly diluted copolymer solutions for analysis, in this case, unreacted collagen remains in the filtrate during filtration, and the copolymer does not pass through the filter with a pore size of 0.45 μm due to the cross-linked structure. A similar result was observed in [36] when studying the MW characteristics of gelatin copolymers.

Further experiments have confirmed this. The analysis of hydrolysates becomes possible because the cross-linked structure is broken in contrast to the copolymer from synthesis. MWD curves of copolymer hydrolysates in the case of pancreatin (Figure 2) and even more so in the presence of thrombin (Figure 3) confirm the formation of a grafted copolymer. When hydrolyzed by pancreatin, the high-molecular mode gradually disappears (Figure 2), and only corresponding low-molecular fractions of collagen remain, which are not hydrolyzed with values of ~20 kDa and ~10 kDa [33,48]. If we compare the obtained results of copolymer hydrolysis with those for collagen [48], then a noticeable decrease in the rate of copolymer hydrolysis occurs, although results are identical in three days. It is well known that proteolytic enzymes break down protein bonds created by arginine and lysine [49]. When synthetic fragments are added to the collagen molecule, they do not affect the breakdown of peptide molecules by those same bonds. Nevertheless, steric difficulties caused by grafted synthetic fragments onto collagen slow down this process. The MWD curves of copolymers hydrolyzed by thrombin do not undergo noticeable changes over time (Figure 3). The inertia of the copolymer to thrombin hydrolysis is because the active center of the globular protein thrombin, unlike pancreatin, is unable to approach the peptide bond of the copolymer.

The morphology of copolymers obtained at different temperatures, shown in Figure 4b–d, varies significantly compared to the morphology of collagen (Figure 4a). The collagen sponge has clear outlines of collagen fibers and formed pores (Figure 4a), and the morphology of the copolymer sponges clearly shows the denser contours of the collagen matrix due to the grafted synthetic fragments of the copolymer (Figure 4b–d). The pore size of the copolymers is different for different temperatures: the copolymer synthesized at 25 °C has a pore size close to collagen ~20 μm (Figure 4b). When the synthesis temperature rises to 45 °C (Figure 4b), the pore size varies from 30 to 100 μm; this can be explained by the disruption of the bonds of individual collagen α-chains and the formation of their chaotic intermolecular interactions due to partial denaturation of the protein in the copolymer. As a result, the resulting cross-linked structure changes. In the copolymer synthesized at 60 °C, the pore size is leveled to ~40 μm (Figure 4d).

The analysis of the literature and the obtained results suggest the formation of a copolymer as follows. The processing of collagen with TBB leads to the formation of borated collagen following the Figure 1:

The release of butane was confirmed earlier by chromatography–mass spectrometry [33]. Borated collagen can initiate the polymer chain addition of MMA in two distinct manners: reversible inhibition (Figure 2) or via the well-known scheme due to the separation of hydrogen atoms by radicals formed during the oxidation of TBB by residual oxygen [50]. The resultant growth radicals interact with *p*-quinone through a radical substitution process on the boron atom, forming a cross-linked polymer structure (Figure 3). Thus, the growth of the grafted chain begins (“grafting from”) and ends (“grafting to”) on collagen.

The cytotoxicity parameter is essential to apply the obtained copolymers as a basis for the scaffolds. Evaluation of cytotoxicity allows for predicting the prospects of synthesized copolymers for use in biomedical purposes. All the copolymers obtained were examined for cytotoxicity using the MTT assay. The data of the analysis results of Sample I—collagen, MMA, TBB, and 2,5-DTBQ (60 °C); Sample II—collagen, MMA, TBB, and 2,5-DTBQ (45 °C); and Sample III—collagen, MMA, TBB, and 2,5-DTBQ (25 °C) are presented in Table 2.

Table 2 shows that the Col–*co*–PMMA samples do not have any cytotoxicity or only have a slight amount. Sample I and its dilutions showed Rank 1 cytotoxicity, with the RGI level remaining consistent at an average of 82.74 ± 1.35%. Sample II also exhibited Rank 1 cytotoxicity in its extract and 1:1 dilution but showed a slight degree of cytotoxicity in subsequent dilutions (1:2–1:8). Conversely, Sample III exhibited a little bit of cytotoxicity (Rank 2) when tested in its 1:1 and 1:2 dilutions. No cytotoxicity was observed in its dilutions of 1:4 and 1:8 (Rank 1). Notably, when examining cells cultured with Samples II and III, a thin film-like layer with wave-like folds was observed in the subcellular layer (Figure 5). It is possible that these copolymers spontaneously form such structures upon contact with cells or their waste products.

Summarizing the data obtained above, we can conclude that from the point of view of the biomedical application, the collagen and PMMA copolymer synthesized at a temperature of 60 °C is the most promising.

Col–*co*–PMMA copolymers were synthesized using the TBB–2,5-DTBQ initiation system with varying *p*-quinone percentages relative to MMA (0.15 mol.%, 0.25 mol.%, 0.35 mol.%, and 2.5 mol.%) to examine the impact of 2,5-DTBQ on the qualities of the copolymer under the current temperature. The amount of unreacted monomer and homopolymer and the proportion of grafted polymer relative to the initial MMA are presented in Table 3. When the concentration of *p*-quinone is higher, more chain transfer acts are likely to occur (Figure 3). This results in the release of butyl radicals into the reaction mixture, which increases the yield of the homopolymer and reduces the proportion of grafted PMMA.

According to the results of the cytotoxicity assessment, Sample IV (a copolymer containing 0.15 mol.% of 2,5-DTBQ) is not toxic. The cytotoxicity rank of the extract and its dilutions varied from 0 to 1 (Table 4). Notably, diluting to 1:8, the RGI was significantly higher than 100%. The latter indicates that low concentrations of copolymer stimulate cell growth. Sample V, which contained 0.35 mol.% of 2,5-DTBQ, showed similar cytotoxicity to Sample I. Nonetheless, when the extract of Sample V was diluted, it increased RGI (Table 4). The results obtained in the study of Sample VI (a copolymer containing 2.5 mol.% of 2,5-DTBQ) differed sharply from the results of Samples I–V. The extract and its 1:1 dilution showed pronounced cytotoxicity corresponding to Rank 3. The toxicity is likely caused by unreacted *p*-quinone. As the extract was diluted, the copolymer became less toxic. At dilutions of 1:4 and 1:8, the toxicity level was Rank 1. However, even at high dilutions, Sample VI still had toxicity, as evidenced by affected cells observed under the microscope (Figure 6).

Thus, Samples I and IV synthesized with low concentrations of *p*-quinone—0.25 mol.% and 0.15 mol.%, respectively—are the most promising for use in scaffold technologies (Figure 7). For further studies, the concentration of *p*-quinone equal to 0.25 mol.% was chosen since it allows to obtain more branched polymer and, at the same time, does not influence the toxicity effect.

To confirm the concepts put forward at a quinone concentration of 0.25 mol.%, Col–*co*–PMMA copolymers were synthesized using the following *p*-quinones: BQ, DQ, and NQ. The amount of unreacted monomer, homopolymer, and the proportion of grafted polymer depends on the structure of *p*-quinone (Table 5). Thus, in the presence of the weakest inhibitors [51]—2,5-DTBQ and DQ—the proportion of grafted PMMA is ~40%; a more potent inhibitor, NQ, leads to a copolymer containing 20% PMMA. An increase in the proportion of grafted PMMA and an insignificant amount of homopolymer is present using the most potent inhibitor (BQ) we associate with the course of controlled polymerization characteristic of the alkylborane–*p*-quinone system [47].

The GPC method was ineffective in analyzing copolymers synthesized with BQ, DQ, and NQ due to the formation of a cross-linked structure, which is similar to copolymers synthesized with 2,5-DTBQ (Figure 1). After the hydrolysis of copolymers by collagenase, the most effective collagen destruction enzyme [52], a water-insoluble polymer, is isolated from the grafted copolymer in a separate phase during the day, which was filtered and analyzed by NMR and GPC methods. ^1^H NMR spectra have shown a wide signal at 3.6 ppm in all samples belonging to the methoxy group of polymethyl methacrylate. The MWD of PMMA (Figure 8) obtained after the hydrolysis of copolymers by collagenase also depends on the inhibitory effect of the *p*-quinone used in the synthesis. The higher the rate constant of the reaction of the growth radical with *p*-quinone, the shorter the chains of the grafted PMMA (Figure 3) and the lower the molecular weight. The inhibitory effect of quinone decreases in the following order: BQ, NQ, 2,5-DTBQ, and DQ [51]. In the same sequence, the molecular weight of the grafted PMMA isolated after collagenase hydrolysis increases (Figure 8). This result provides further evidence that PMMA was indeed grafted onto collagen.

The morphology of copolymers using the next *p*-quinones—BQ, DQ, and NQ (Figure 9b–d), similar to the copolymer synthesized in the presence of 2,5-DTBQ (Figure 4d)—differs only in the pore size. So, the pore size in the case of 2,5-DTBQ is ~40 μm (Figure 4d), BQ is ~20 μm, DQ is ~65 μm, and NQ is ~30 μm. The pore sizes are in strict accordance with the inhibitory effect of *p*-quinones [51]: the shorter the polymer chain of the grafted PMMA (Figure 3), the shorter the “cross-linking” and the smaller the pore size of the freeze-dried sample. For PMMA synthesized in the presence of the alkylborane–*p*-quinone system, there is a strict correspondence of the molecular weights of polymers with the inhibitory action of *p*-quinones [47].

After analyzing Samples VII–IX (Col-*co*-PMMA in the presence of TBB–*p*-quinone system using NQ (Sample VII), DQ (Sample VIII), and BQ (Sample IX)), it was discovered that the extracts had some level of toxicity. However, upon dilution, the degree of toxicity varied in different ways (Table 6). Based on the results, Sample VII was Rank 3 for cytotoxicity and remained cytotoxic even after being diluted. Sample VIII had a lower level of cytotoxicity, Rank 2, and the cytotoxic effect decreased to Rank 1 after dilution (Table 6). Sample IX was ranked 2 for cytotoxicity, and a dilution of 1:1—Rank 3. It is worth noting that the RGI values for the extract and its 1:1 dilution were close to the boundary between Ranks 2 and 3. However, with further dilution, the cytotoxic effect of the copolymer decreased until it reached 0. When diluted at ratios of 1:4 or 1:8, the RGI showed an over 100% increase in cell proliferation due to the low content of copolymer.

## 4. Conclusions

Thus, polymerization of MMA in the aqueous dispersion of collagen in the presence of tri-*n*-butyl borane and *p*-quinone (2,5-DTBQ, BQ, DQ, and NQ) leads to the formation of a grafted cross-linked copolymer of collagen and poly(methyl methacrylate). The quantity of unreacted monomer, homopolymer, and the proportion of grafted PMMA depends on the inhibitory effect of *p*-quinone, which is consistent with the data obtained for (co)polymers of alkyl(meth)acrylates and styrene in the presence of the alkylborane–*p*-quinone system. These copolymers tend to biodegrade when exposed to certain enzymes, are non-toxic, and promote the growth of cells. At the same time, collagen denaturation occurring at high temperatures does not worsen the obtained characteristics. Based on the results, we can present our research as a chemical model of scaffolds. By comparing the copolymer characteristics with the composition of the polymerizing mixture and synthesis conditions, we determined that the optimal method for synthesizing the scaffold precursor involves creating a collagen and PMMA copolymer at 60 °C in a 1% acetic acid dispersion of fish collagen, with a mass ratio of components: collagen:MMA:TBB:2,5-DTBQ equal to 1:1:0.015:0.25.

## Data Availability

The data that support the findings of this study are available on request from the corresponding author.

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
