# Peer review of "Scaffold Chemical Model Based on Collagen—Methyl Methacrylate Graft Copolymers"

_polymers, 2023, doi:10.3390/polym15122618_

Round 1

Reviewer 1 Report

This paper provides a chemical model that fits several criteria for tissue engineering purposes: low cytotoxicity, biodegradable, and biocompatible. The overall research area is significant because cells cultured by such scaffolds can reveal features mimicking a tissue, and so such scaffolds may be used for tissue engineering purpose in a long run. In this paper, the authors develop some polymer-based scaffolds optimal for HDF cell cultures by doping TBB and DTBQ into the collagen:MMA mixture. Overall, some control data are missing, and it is unclear if the morphology mimics the natural tissue. The author should justify these points and include some essential data before publication.

Major comments:

1.    To help readers go through the article, the authors should clearly define terms in early paragraphs and use the terms consistently. For instance, it’s unclear if “graft” is interchangeable to other words used in the article: “combine”, “copolymerize”, or “crosslink.” “Unreacted” means degraded or unpolymerized depending on context, which confuses the readers. Furthermore, the idea between “grafting to” and “grafting from” is not properly defined in the Introduction section.

2.    For Control data

a.    In Figure 1, the temperature of the condition 1 is not specified. Furthermore, the authors should include collagen at varying temperatures associated with the conditions 2-4. All peaks need to be explained in the text (e.g. degraded or polymers).

b.    In Figures 5 and 6, the authors need to show the representative images for the optimal conditions, as mentioned in Lines 337-338 and 368-370.

c.     Why do the authors try the condition above 45C if collagen starts denaturation? Tissue architecture can be important for cell and tissue physiology (PMID: 30146160), and denaturation may lead to some unwanted problems.

3.    As authors specified in the Introduction, one important prerequisite for scaffold is to help the tissue develop proper morphology. However, this component is missing in the paper. Furthermore, it is unclear if the assay supports the argument of increased proliferation.

4.    It is unclear how authors define pore size, and this approach is not included in the Methods. In principle, the authors can consider particle tracking and mean-square-displacement methods commonly used for colloidal sciences to define pore size accurately.

Minor comments:

1.    Here are several suggestions for data presentation.

a.    As there are spaces in Figures 1, 2 and 7, the authors may consider putting experimental conditions in figures instead of numbers.

b.    For time series in Figure 2, the authors may consider gray colors with different shades, so that the readers can easily see if each peak increases or decreases along with time. Furthermore, the authors should plot intensities vs. time for the lgM peaks around 4 and 5.5 because their slopes reveal the reaction kinetics.

2.    Scale bars in Figures 4 and 8 are not properly addressed.

3.    In the Methods section, it’s unclear what the Mg chip is.

4.    The sentence in Lines 63-64 is confusing.

5.    In Line 293, “Fig. 4d”

The presentation of the diagrams need to be improved, as current setups can dilute the messages. Furthermore, the definition of the used terms needs to be consistent as well.

Author Response

Hello! Response to comments in the attached file.

Reviewer 2 Report

In this manuscript, the authors described the synthesis of collagen-methyl methacrylate copolymers as a scaffold chemical model. Several characterization methods were applied to study and reveal the chemical structure of the resulting copolymers. In addition, the authors displayed that the products were biodegradable in the presence of enzymes, non-toxic, and could stimulate cell growth. Overall, this manuscript is systematic and comprehensive, which could be accepted after a minor revision. Please address the following questions and comments.

1. In previous work, the authors reported a very similar system on gelatin (Ref 38). Those two works share a great amount of similarities, from concept to experimental design, which largely lower the novelty of this manuscript. Could the authors explain the reason to study collagen-based system? What are the benefits and improvements of the collage-based system compared to gelatin-based one?

2. One suggestion on Figure 1, 2, and 3 for a better presentation, could the authors include the sample information for each colored line in the Figure (for example, right next to the line)? It is not easy to read the data by jumping from the caption to the figure back and forth.

3. Based on the information provided by the authors, the collagen solution was prepared in an aqueous solution with acetic acid. Could the water and acetic acid react with organoborane and radicals during the polymerization?

4. More characterization methods could be included to analyze the cross-linked structure, such as FTIR and XPS.

Some sentences could be polished in a better way.

Author Response

(The authors gave the same response as above.)

Round 2

Reviewer 1 Report

My comments are addressed properly.